# Dual Antiplatelet Therapy: A Concise Review for Clinicians

**DOI:** 10.3390/life13071580

**Published:** 2023-07-18

**Authors:** Hafeez Ul Hassan Virk, Johao Escobar, Mario Rodriguez, Eric R. Bates, Umair Khalid, Hani Jneid, Yochai Birnbaum, Glenn N. Levine, Sidney C. Smith, Chayakrit Krittanawong

**Affiliations:** 1Harrington Heart & Vascular Institute, Case Western Reserve University, University Hospitals Cleveland Medical Center, Cleveland, OH 44101, USA; 2International Transitional Medical Graduate, American College of Physician, Philadelphia, PA 19106, USA; 3John T Milliken Department of Medicine, Division of Cardiovascular Disease, Section of Advanced Heart Failure and Transplant, Barnes-Jewish Hospital, Washington University, St. Louis School of Medicine, St. Louis, MO 63110, USA; 4Division of Cardiovascular Medicine, Department of Internal Medicine, University of Michigan, Ann Arbor, MI 48109, USA; 5Michael E. DeBakey VA Medical Center, Section of Cardiology, Baylor College of Medicine, Houston, TX 77030, USA; 6Division of Cardiology, University of Texas Medical Branch, Houston, TX 77555, USA; 7Division of Cardiology, McAllister Heart Institute, University of North Carolina at Chapel Hill, Chapel Hill, NC 27599, USA; 8Cardiology Division, NYU School of Medicine, NYU Langone Health, New York, NY 10016, USA

**Keywords:** dual antiplatelet therapy, DAPT

## Abstract

Dual antiplatelet therapy (DAPT) combines two antiplatelet agents to decrease the risk of thrombotic complications associated with atherosclerotic cardiovascular diseases. Emerging data about the duration of DAPT is being published continuously. New approaches are trying to balance the time, benefits, and risks for patients taking DAPT for established cardiovascular diseases. Short-term dual DAPT of 3–6 months, or even 1 month in high-bleeding risk patients, is equivalent in terms of efficacy and effectiveness compared to long-term DAPT for patients who experienced percutaneous coronary intervention in an acute coronary syndrome setting. Prolonged DAPT beyond 12 months reduces stent thrombosis, major adverse cardiovascular events, and myocardial infarction rates but increases bleeding risk. Extended DAPT does not significantly benefit stable coronary artery disease patients in reducing stroke, myocardial infarction, or cardiovascular death. Ticagrelor and aspirin reduce cardiovascular events in stable coronary artery disease with diabetes but carry a higher bleeding risk. Antiplatelet therapy duration in atrial fibrillation patients after percutaneous coronary intervention depends on individual characteristics and bleeding risk. Antiplatelet therapy is crucial for post-coronary artery bypass graft and transcatheter aortic valve implantation; Aspirin (ASA) monotherapy is preferred. Antiplatelet therapy duration in peripheral artery disease depends on the scenario. Adding vorapaxar and cilostazol may benefit secondary prevention and claudication, respectively. Carotid artery disease patients with transient ischemic attack or stroke benefit from antiplatelet therapy and combining ASA and clopidogrel is more effective than ASA alone. The optimal duration of DAPT after carotid artery stenting is uncertain. Resistance to ASA and clopidogrel poses an incremental risk of deleterious cardiovascular events and stroke. The selection and duration of antiplatelet therapy in patients with cardiovascular disease requires careful consideration of both efficacy and safety outcomes. The use of combination therapies may provide added benefits but should be weighed against the risk of bleeding. Further research and clinical trials are needed to optimize antiplatelet treatment in different patient populations and clinical scenarios.

## 1. Introduction

Dual antiplatelet therapy (DAPT) combines two antiplatelet drugs, typically aspirin (ASA) and a P2Y12 inhibitor, to treat atherosclerotic cardiovascular disease. DAPT aims to inhibit platelet activation and aggregation, reducing the risk of thrombotic events [1]. Despite its widespread use, the optimal duration, specific drug combinations, and potential bleeding risks associated with DAPT remain subjects of ongoing debate and investigation.

One of the most fearful side effects of antiplatelet therapy is bleeding [2]. Therefore, balancing ischemic and bleeding risks is key when deciding to initiate antiplatelet treatment. Managing bleeding risk is inherently linked to the treatment decisions made throughout patients’ medical history with antiplatelet therapy indications. Patients with high bleeding risk are those with previous conditions such as elderly, stroke, liver or renal dysfunction, prior bleeding, trauma, cancer, anemia, thrombocytopenia, and the use of medications that affect the hemostasis process [2,3,4,5]. Timely decisions on the duration of antiplatelet therapy can strongly impact the patient’s prognosis while treating their established cardiovascular conditions [2].

This review will provide a comprehensive overview of the current DAPT evidence base, including indications, efficacy, safety, and optimal duration of therapy in many cardiovascular contexts in which DAPT could be utilized.

## 2. Overview of the Antiplatelet Therapies 

Historically, ASA has been the most commonly prescribed antiplatelet therapy as monotherapy or combination therapy for decades [6]. However, the development of other oral antiplatelet medications, such as indobufen, clopidogrel, prasugrel, ticagrelor, dipyridamole, cilostazol, and vorapaxar, has broadened the treatment options for patients with cardiovascular diseases (Figure 1, Table 1).

### 2.1. Aspirin

ASA has been proven to have various benefits in decreasing the cardiovascular event rate in patients with prior myocardial infarction (MI), stroke, or PAD [7]. ASA functions by irreversibly inhibiting the cyclooxygenase (COX) 1 and 2 enzymes involved in the thromboxane A2 production, a potent platelet aggregator [8,9]. 

### 2.2. Indobufen

Indobufen, an antiplatelet agent used in Europe, has effectively reduced the risk of recurrent strokes [10]. Indobufen is a reversible inhibitor of COX and has similar effects to aspirin [11].

The OPTION study, a non-inferiority trial, compared indobufen-based DAPT and conventional DAPT in patients with coronary stents. This trial aimed to determine if there was a significant difference in the composite of stroke, stent thrombosis, nonfatal myocardial infarction, cardiovascular death, or bleeding events in one year among the treatment groups. The indobufen-based DAPT showed a primary endpoint event of 4.47% compared to 6.11% in the conventional DAPT group [12]. Additionally, a lower incidence of bleeding was seen in the indobufen-based DAPT group. Indobufen is already approved in China and Europe for treating atherosclerotic cardiovascular disease but is not currently approved for use in the United States.

### 2.3. P2Y12 Receptor Inhibitors

Clopidogrel, prasugrel, and ticagrelor are platelet P2Y12 receptor inhibitors. These drugs have varying degrees of potency and onset of action. Prasugrel and ticagrelor have faster onset, higher potency, and more predictable platelet inhibition than clopidogrel.

### 2.4. Clopidogrel

Clopidogrel effectively reduces the risk of cardiovascular events and recurrent stroke compared with aspirin, although at an increased bleeding risk [13]. It is a prodrug that has variable activation depending upon genetic loss-of-function alleles. Clopidogrel is often combined with aspirin in patients after elective percutaneous coronary intervention (PCI) [14].

### 2.5. Prasugrel

Prasugrel is another prodrug that requires activation before binding to the platelet P2Y12 receptor to inhibit platelet aggregation [15]. In comparison to clopidogrel, prasugrel exhibits greater potency, faster onset of action, and more efficient metabolite activation [16]. Prasugrel has significantly reduced cardiovascular events, such as stent thrombosis, although it carries a higher risk of major bleeding than clopidogrel. However, there is no significant difference in overall mortality among prasugrel and clopidogrel [17]. 

### 2.6. Ticagrelor

Ticagrelor, a non-prodrug, binds reversibly to the adenosine diphosphate (ADP) P2Y12 receptor, inhibiting platelets without demanding metabolic activation [18]. It has a shorter half-life and needs to be taken twice daily. Compared to clopidogrel, ticagrelor achieves faster platelet inhibition [16]. Ticagrelor protects against ischemia-reperfusion injury (IRI) during acute MI and long-term treatment. Moreover, it reduces harmful inflammation, prevents adverse cardiac remodeling and atherosclerosis, and enhances stem cell recruitment. These positive effects are attributed to ticagrelor’s ability to increase adenosine levels, activating protective molecules in the heart’s affected area [19].

The PLATO study established that ACS patients receiving ticagrelor for 12 months had fewer MIs, strokes, or cardiovascular deaths than clopidogrel. Ticagrelor had similar bleeding risk but more dyspnea and ventricular pauses compared to clopidogrel [20]. According to the ATLANTIC study, the use of ticagrelor before hospital arrival did not lead to enhanced coronary reperfusion before PCI in patients with ACS compared to the in-hospital ticagrelor administration. However, ticagrelor was safe in both pre-hospital and in-hospital settings. Although more evidence is needed to evaluate ticagrelor with other antiplatelet agents, current data suggests that ticagrelor is a valuable alternative for precluding thrombotic cardiovascular events in ACS patients, regardless of whether they undergo invasive or noninvasive management [21]. 

### 2.7. Prasugrel vs. Ticagrelor

An unexpected finding emerged in the ISAR-REACT 5 study, which was a multicenter, randomized, open-label trial. This study revealed that prasugrel had a lower incidence of cardiovascular death, MI, or stroke compared to ticagrelor in patients with ACS [22]. Moreover, the study revealed that ticagrelor was linked with a notable increase in the chance of recurrent MI than prasugrel [23]. 

One study found a reduced risk of bleeding with prasugrel compared to ticagrelor in ACS patients following PCI [24]. On the other hand, the SWEDEHEART registry noted no statistical differences between ticagrelor and prasugrel regarding efficacy and safety in ACS patients undergoing PCI [25]. 

### 2.8. Dipyridamole

Dipyridamole inhibits platelet phosphodiesterase, reducing platelet aggregation and increasing levels of cyclic adenosine monophosphate (cAMP) and cyclic guanine monophosphate (cGMP). It also inhibits adenosine reuptake, leading to higher interstitial adenosine levels. Adenosine receptor activation increases cAMP levels [26]. Dipyridamole can also cause vasodilation [27]. A meta-analysis indicated that ASA plus dipyridamole significantly reduced stroke risk compared to ASA alone (RR 0.77, 95% CI 0.67–0.89) [28]. In another meta-analysis, ASA alone or combined with dipyridamole had no significant effect on the incidence of a composite outcome of nonfatal stroke, nonfatal MI, and cardiovascular death but reduced nonfatal stroke risk (RR 0.66; 95% CI 0.47–0.94) [29]. 

### 2.9. Cilostazol

Cilostazol is a by-product of 2-oxy quinolone that interferes with phosphodiesterase III activity, promoting elevated levels of cAMP [30]. Cilostazol prevents cerebral infarction and increases exercise time in patients with claudication [31,32,33,34].

### 2.10. Vorapaxar 

Vorapaxar is a medication that acts as an antagonist for protease-activated receptor-1 (PAR-1), effectively blocking platelet activation induced by thrombin. Clinical studies demonstrated that Vorapaxar could lower the risk of cardiovascular events in individuals with previous PAD or MI when used alone or combined with standard antiplatelet treatment. However, it’s important to note that Vorapaxar is associated with a high risk of bleeding, especially in individuals with a prior transient ischemic attack or stroke [35,36]. Nonetheless, the TRACER trial reveals that adding vorapaxar to the standard therapy in ACS patients did not decrease the primary combined end-point of death from stroke, MI, or cardiovascular etiologies [37]. Although Food and Drug Administration approved, the drug is not commonly used in clinical practice [38].

## 3. Duration of Antiplatelet Therapy in ACS 

### 3.1. Evidence for Short-Term Dual Antiplatelet Therapy 

Since DAPT was utilized initially for a period of 12 months for ACS and the prevention of drug-eluting stent thrombosis, short-term DAPT generally refers to a duration of therapy shorter than this, often for 3–6 months, and some more recent studies suggest just 1 month in high bleeding risk patients [39,40]. The short-term DAPT safety and efficacy in people with ACS have been previously assessed (Figure 2). The CURE trial showed that combining clopidogrel and aspirin for 3 to 12 months in ACS patients with NSTEMI decreased myocardial infarction events, ischemic recurrence, stroke, and cardiovascular mortality with increased bleeding risk [14]. 

In meta-analyses that investigated the optimal duration of DAPT among patients undergoing PCI, short-term DAPT is supported in the setting of the newer generation drug-eluting stents (DES), especially in elderly patients and non-east Asians [41,42,43]. One meta-analysis showed that short-term DAPT (<6 months) following PCI for the ACS treatment was not related to an augmented threat of stent thrombosis than long-term DAPT [39]. The REDUCE trial found no difference in outcomes between 3- and 12-month DAPT in both genders at one-year and two-year follow-ups, including mortality, MI, stroke, stent thrombosis occurrence, revascularization, and bleeding rates [44]. 

In the SMART-CHOICE trial, individuals undergoing PCI received three months of DAPT, and they were divided into two groups: continue DATP or take a PY12 inhibitor as a single therapy for nine months more. At 12 months, major adverse events were similar, but the bleeding was lower in the P2Y12 inhibitor group [45].

The management of non-ST elevation ACS, guided by the 2014 American College of Cardiology (ACC)/American Heart Association (AHA) guidelines and the 2020 European Society of Cardiology (ESC) guidelines, suggests either clopidogrel, prasugrel, or ticagrelor for initial “up-front” therapy, regardless of planned treatment. Ticagrelor and prasugrel are favored over clopidogrel, but prasugrel is primarily recommended for individuals who do not have an increased risk of bleeding and are scheduled for PCI [46,47].

### 3.2. Evidence for Extended-Duration Dual Antiplatelet Therapy 

Extended duration DAPT is generally defined as DAPT >12 months. Extended duration of DAPT in managing ACS involves a trade-off between decreased ischemic and increased bleeding complications. The use of extended-duration DAPT has been related to lower incidence of major cardiovascular events with an increased risk of bleeding. Nevertheless, the precise length of DAPT remains to be determined since most studies investigating prolonged DAPT have only observed patients for a limited period, despite treating them for several years (as no trial can realistically be continued “indefinitely”).

The DAPT trial studied 9961 patients who received a drug-eluting stent after a coronary procedure. After 12 months of thienopyridine drug treatment and aspirin, patients without complications were randomly assigned to continue the P2Y12 inhibitor or receive a placebo for 18 months while all took aspirin. Thienopyridine reduced stent thrombosis, adverse cardiovascular events, and MI compared to placebo but had a higher death rate and increased bleeding risk. Both groups had an increased risk of stent thrombosis and MI after stopping thienopyridine [48].

In a recent study called STOPDAPT-2 ACS, researchers examined ACS patients who underwent a successful PCI. The study compared two treatment approaches: DAPT for 1 to 2 months, followed by clopidogrel monotherapy, and standard 12 months of DAPT with aspirin (ASA) and clopidogrel. The findings revealed that clopidogrel monotherapy after 1 to 2 months of DAPT in patients with increased ischemic risk is not inferior to the standard 12 months of DAPT in terms of clinical benefits. Interestingly, although there was a reduction in bleeding events, the use of clopidogrel monotherapy was associated with increased cardiovascular events [40]. 

The HOST-EXAM trial, a prospective, randomized, open-label, multicenter trial, examined individuals aged ≥ 20 years who received DAPT for 6–18 months following PCI using DES. A subset of patients was on clopidogrel 75 mg once daily and the other group was taking aspirin 100 mg per day for 24 months. The clopidogrel subset had lower occurrence (5.7%) of a composite of MI, stroke, readmission because of ACS, all-cause death, and Bleeding Academic Research Consortium type ≥3, compared to the aspirin group (7.7%) (hazard ratio 0.73 [95% CI 0.59–0.90]; *p* = 0.0035) [49]. 

## 4. Duration of Antiplatelet Therapy in Stable CAD

In the CHARISMA trial, individuals with either clinically evident cardiovascular disease or multiple risk factors were randomly divided into two groups. One group received a combination of clopidogrel (75 mg per day) and low-dose aspirin (75 to 162 mg per day), while the other group received a placebo along with low-dose aspirin. This study did not reveal any substantial advantages of extended DAPT in terms of reducing the incidence of stroke, MI, or cardiovascular fatalities, with a median duration of 28 months. Further, the study indicated potential disadvantages in patients with several risk factors, such as bleeding [50].

The THEMIS trial studied ticagrelor plus ASA versus placebo plus ASA in 19,220 patients with type 2 diabetes and stable coronary artery disease. No previous history of MI or stroke was present. The median follow-up was 39.9 months. The ticagrelor group (7.7%) had a lower composite endpoint (CV death, MI, or stroke) compared to the placebo group (8.5%) with a hazard ratio of 0.90 (95% CI 0.81–0.99, *p* = 0.04). Fatal bleeding did not significantly differ [51]. More importantly, the THEMIS PCI trial focused on patients with prior PCI. The ticagrelor group had a lower primary endpoint rate (7.3%), which was a combination of CV death, MI, or stroke compared to placebo (8.6%), with a hazard ratio of 0.85 (CI 0.74–0.97, *p* = 0.013). Ticagrelor had higher TIMI major bleeding incidence (2% vs. 1.1%) with a hazard ratio of 2.03 (CI 1.48–2.76, *p* < 0.0001) [52].

The GLOBAL-LEADERS trial studied CAD or ACS patients undergoing PCI treated for two years with one specific antiplatelet approach. First, patients received DAPT with aspirin and ticagrelor for one month. In the following 23 months, patients were assigned to continue DAPT with aspirin and ticagrelor, take ticagrelor alone, or receive standard treatment with aspirin plus clopidogrel. Ticagrelor did not show superiority over standard therapy in preventing all-cause mortality or new Q-wave MI. Bleeding rates were similar in both groups [53]. 

The COMPASS trial determined that the combination of rivaroxaban plus aspirin was more efficacious than aspirin alone in reducing death, stroke, and myocardial infarction by 24% in patients with stable CAD. Still, a substantial increase in the bleeding rate was observed with this approach [54]. 

## 5. Duration of Antiplatelet Therapy in Post-PCI with Atrial Fibrillation in Addition to Anticoagulation

Several guidelines recommend the duration of antiplatelet therapy in atrial fibrillation (AF) patients after PCI. According to the 2018 ESC guidelines on DAPT duration in patients taking oral anticoagulation, such as those with AF, triple therapy (aspirin, clopidogrel, and an oral anticoagulant) should be considered for 1 month independently of the stent utilized in the PCI. In patients with increased ischemic risk, prolonging the triple therapy up to 6 months is suggested since the benefits of this approach justify the risk of bleeding. However, if the bleeding has a great probability of occurrence, a combination of clopidogrel and oral anticoagulant is an option to treat these patients in the first month after PCI. After 12 months, these guidelines favor withdrawing the antiplatelet therapy and keeping the oral anticoagulant based on the patient’s needs [55]. 

The 2021 AHA/ACC/SCAI Revascularization Guidelines recommend that patients with AF who have experienced PCI should discontinue ASA treatment no later than 4 weeks after PCI. Then, these patients should continue the P2Y12 inhibitor in combination with a non-vitamin K oral anticoagulant (apixaban, rivaroxaban, edoxaban, or dabigatran) to reduce the risk of bleeding [56]. 

Risk stratification tools, such as CHA2DS2-VASc and HAS-BLED scores, have been designed to help identify patients with AF at a higher risk of adverse events following PCI and guide decision-making regarding antiplatelet therapy duration [57].

## 6. Duration of Antiplatelet Therapy in CABG

Current guidelines recommend ASA should be started promptly in patients undergoing coronary artery bypass surgery (CABG), ideally in the following six hours after completing the procedure at 100–325 mg. Then, ASA should be kept indefinitely to decrease the risk of saphenous vein graft closure and adverse cardiovascular events. Furthermore, the benefits of DAPT therapy in CABG patients, either ASA + clopidogrel or ticagrelor, for 1 year are associated with enhanced graft patency in comparison to ASA monotherapy [56]. 

## 7. Duration of Antiplatelet Therapy Post TAVI

Transcatheter aortic valve implantation (TAVI) is a promising option to surgical approach in high-risk individuals or those ineligible for surgery [58]. Nonetheless, as with any medical procedure, TAVI comes with its own set of risks and potential complications, including bleeding and stroke.

After TAVI, the ideal duration of antiplatelet therapy is still being determined. It is advisable to use only ASA or clopidogrel for patients needing anticoagulation during the periprocedural period. Anticoagulation therapy should be given with or without a single antiplatelet therapy for 3–6 months, depending on bleeding risk. Afterward, it is recommended to switch to oral anticoagulation monotherapy. The use of ASA or clopidogrel during the periprocedural period is suggested for patients not needing anticoagulation. In the first 3–6 months, the decision between monotherapy or the DAPT approach should be based on the bleeding risk. After this period, it is suggested to continue with monotherapy [59].

One meta-analysis compared the use of antiplatelet therapy after TAVI and found that using ASA monotherapy resulted in lower bleeding rates without increasing the risk of stroke or death, as compared to using DAPT with either 3- or 6-month treatment duration [60]. Interestingly, clopidogrel is more effective than aspirin in preventing mortality from cardiovascular causes after 24 months of therapy, independently of anticoagulant utilization [61].

## 8. Duration of Antiplatelet Therapy in PAD 

PAD is a common vascular disease that affects millions of people worldwide. One meta-analysis concluded that the clinical benefit of antiplatelet monotherapy is reduced in asymptomatic PAD and only slightly beneficial for symptomatic conditions. However, PAD is associated with an apparent risk of major bleeding [62].

Notably, in individuals with prior MI and PAD without a previous history of stroke or TIA, vorapaxar decreased the ischemic limb event by 42% and the necessity of peripheral revascularization by 16% in patients with intermittent claudication. Nonetheless, the incidence of bleeding was 62% higher with vorapaxar compared to placebo [63].

Various meta-analyses have shown that cilostazol is effective in treating stable moderate-to-severe claudication [64,65,66,67]. In addition, four trials showed that antiplatelet treatment (piconamide during 18 months, ticlopidine during 6–21 months) lowered the risk of revascularization when compared to placebo [68,69,70,71]. 

In the THEMIS trial, major adverse limb events happened less frequently with ticagrelor/aspirin (1.3%) than with placebo/aspirin (1.6%). Peripheral revascularization and acute limb ischemia also exhibited lower rates in the ticagrelor/aspirin group. The primary outcome of irreversible harm, including death, MI, stroke, bleeding, or intracranial hemorrhage, was lower with ticagrelor/aspirin (9.3%) than with placebo/aspirin (11.0%) [51]. 

## 9. Duration of Antiplatelet Therapy in PAD Post-Peripheral Stent

While the duration of DAPT after coronary stenting has been extensively studied and standardized, the optimal time in the context of PAD post-peripheral stent still needs to be clarified. Balancing the risk of stent thrombosis against the potential for bleeding complications poses a unique challenge in this patient population due to the peripheral vascular anatomy, comorbidities, and inherent bleeding risks associated with PAD [72].

According to the ESC guidelines, it is recommended to administer DAPT for a minimum of one month following endovascular revascularization, regardless of the type of stent used (bare metal or drug-eluting) [73]. One study showed that revascularized PAD individuals treated with clopidogrel in combination with aspirin were associated with fewer revascularization events compared to aspirin plus placebo in the first six months of therapy [74]. In the MIRROR trial, PAD individuals receiving DAPT showed less platelet activation after revascularization and fewer reinterventions than patients in the aspirin plus placebo group. Additionally, DAPT among these patients did not increase bleeding [75]. 

## 10. Duration of Antiplatelet Therapy in Carotid Artery Disease

Aspirin or clopidogrel are commonly used to decrease the occurrence of cerebrovascular disease in patients with carotid disease [76]. Insufficient proof exists to support the notion that antiplatelet therapy is effective in precluding strokes in individuals without symptoms [77]. Nevertheless, data suggests that the combination of ASA and clopidogrel during 7 days in patients with carotid disease is more beneficial than ASA alone in lowering microembolization signals at day 2 in patients with symptomatic intracranial stenosis (ISS) [78]. In asymptomatic carotid artery disease, the ESC guidelines recommend a single antiplatelet therapy for at least 1 year [73]. Aspirin + dipyridamole is better than aspirin alone at preventing recurrent cerebrovascular event in patients with transient ischemic attack (TIA) or ischemic stroke. This therapy can be started within 24 h of symptoms. Long-term aspirin + dipyridamole is not better than clopidogrel alone for ischemic stroke or confirmed TIAs with neuroimaging [79].

## 11. Duration of Antiplatelet Therapy in Carotid Artery Disease Post-Carotid Stent

ESC guidelines recommend DAPT for carotid artery stenting (CAS). This recommendation is supported by two trials comparing aspirin alone and DAPT for CAS. These studies were stopped early because of the high occurrence of stent thrombosis and neurological complications in the aspirin monotherapy group [80,81]. These events occurred within 30 days of the procedure and were related to the procedure. 

The optimal duration of DAPT after CAS is still a debate. Tardily brain lesions in magnetic resonance imaging following CAS have been observed after one month, questioning the need for extended DAPT beyond this period. Nevertheless, prolonged DAPT carries risks of hemorrhagic conversion of these lesions leading to intracranial hemorrhage. Patients with recent MI (<12 months) and low bleeding risk may benefit from prolonged DAPT beyond one month after CAS [73].

## 12. Duration of Antiplatelet Therapy in Myocardial Infarction with Non-Obstructive Coronary Arteries 

MINOCA, or myocardial infarction with non-obstructive coronary arteries, refers to various conditions with varying underlying causes. MINOCA is identified by clinical signs indicating a MI, while angiography reveals normal or nearly normal coronary arteries with stenosis of less than 50% [82]. Individuals diagnosed with MINOCA have a reduced likelihood of experiencing recurring cardiovascular events compared to those with MI and obstructive CAD [83]. The majority of trials utilizing DAPT showed no benefits regarding MACE and mortality reduction in patients with MINOCA in the first year [84,85,86,87]. Further studies will be needed to determine the duration of DAPT in patients diagnosed with MINOCA. 

## 13. Duration of Antiplatelet Therapy in Spontaneous Coronary Artery Dissection

Spontaneous coronary artery dissection (SCAD) is characterized by the detachment of the layers within the arterial wall of the epicardial coronary artery due to internal bleeding, with or without a tear in the inner lining [88]. Most SCAD patients treated conservatively receive DAPT in an acute setting and are then discharged with a SAPT to reduce MACE [89,90,91]. However, individuals who undergo stenting are advised to undergo dual antiplatelet therapy for 12 months, followed by a prolonged or lifelong single-drug regimen commonly involving aspirin [90]. Others refrain from using DAPT or only use it for a short period (1–3 months) before transitioning to more extended aspirin treatment [92]. 

## 14. Resistance to Antiplatelet Agents

Clinical antiplatelet resistance refers to the inability to prevent stroke or major vascular events despite the optimal use of antiplatelet drugs [93]. Treatment ASA resistance specifically refers to cardiovascular events occurring even with regular recommended doses of aspirin [94]. Aspirin resistance independently predicts poor outcomes [95]. In a study, patients classified as ASA-resistant had a significantly higher rate of death, MI, or stroke (24% vs. 10%) [96]. 

Clopidogrel’s non-responsiveness refers to adverse events during clopidogrel therapy [97]. High residual platelet reactivity (HRPR) occurs in 16–50% of patients on clopidogrel [98]. One study found resistance in 31% at day five, decreasing to 15% by day 30 [99]. 

## 15. Conclusions

Antiplatelet therapy offers personalized treatment options for cardiovascular diseases. ASA has been the standard therapy, but newer agents such as indobufen, clopidogrel, prasugrel, ticagrelor, dipyridamole, cilostazol, and vorapaxar have expanded treatment options for patients with varying levels of efficacy and safety. Duration and combination should consider patient’s clinical features, medical history, and bleeding risk. The optimal time is uncertain in various procedures and conditions.

The use of DAPT less than 12 months reduces cardiovascular events in ACS. Extended-duration DAPT may reduce cardiovascular events but increases bleeding risks. Optimal duration post-PCI and in AF remains controversial. For CABG patients, ASA is recommended postoperatively to reduce thrombotic events and improve graft patency. DAPT with clopidogrel or ticagrelor and ASA has also shown benefits but comes with a higher risk of bleeding. The optimal duration of DAPT following TAVI remains uncertain. Antiplatelet monotherapy is slightly beneficial in symptomatic patients with PAD compared to asymptomatic disease. Vorapaxar may provide a positive benefit-risk profile in secondary prevention. Antiplatelet therapy can improve pain free walking distance and lower the risk of revascularization in PAD. Combining ASA and clopidogrel decreases microembolization signals in individuals with intracranial symptomatic stenosis. DAPT is recommended over aspirin alone for patients undergoing CAS. ASA resistance and clopidogrel non-responsiveness are predictors of unfavorable clinical outcomes.

## Figures and Tables

**Figure 1 life-13-01580-f001:**
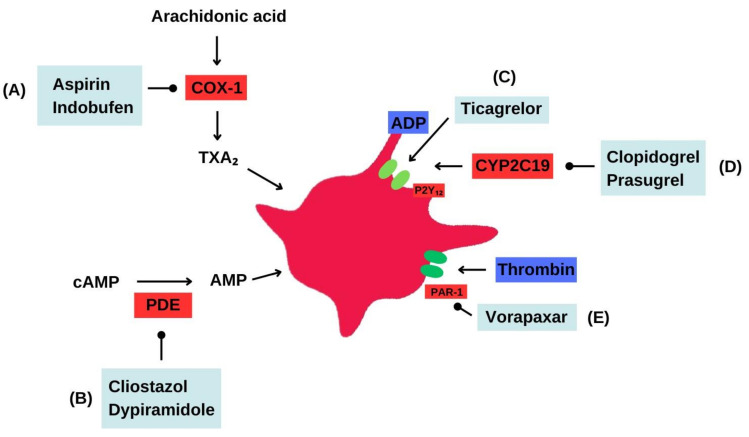
Antiplatelet therapy. Mechanisms of action. (**A**) Aspirin irreversibly inhibits the enzyme cyclooxygenase-1 (COX-1), reducing the formation of thromboxane A2 and effectively inhibiting platelet aggregation. Indobufen also inhibits COX-1, but as a reversible inhibitor. (**B**) Cilostazol inhibits phosphodiesterase type 3 (PDE3), leading to increased levels of intracellular cyclic adenosine 3′,5′-monophosphate (cAMP) in platelets and vascular smooth muscle cells. This results in the inhibition of platelet aggregation and peripheral arterial vasodilation. Dipyridamole, on the other hand, acts as a phosphodiesterase inhibitor, increasing cAMP levels and exerting vasodilatory and antiplatelet effects. Additionally, dipyridamole inhibits the uptake of adenosine by platelets and other tissues, leading to increased extracellular adenosine concentration. (**C**) Ticagrelor directly acts on the P2Y12 receptor as a reversible inhibitor, preventing platelet aggregation and receptor-mediated activation. (**D**) Clopidogrel blocks the P2Y12 receptor on the surface of platelets, thereby inhibiting platelet aggregation and P2Y12-mediated activation. Prasugrel selectively inhibits the P2Y12 receptor on platelets, effectively blocking platelet aggregation and reducing the formation of blood clots. (**E**) Vorapaxar functions as an inhibitor of the protease-activated receptor-1 (PAR-1), reducing platelet response to thrombin and consequently decreasing blood clot formation.

**Figure 2 life-13-01580-f002:**
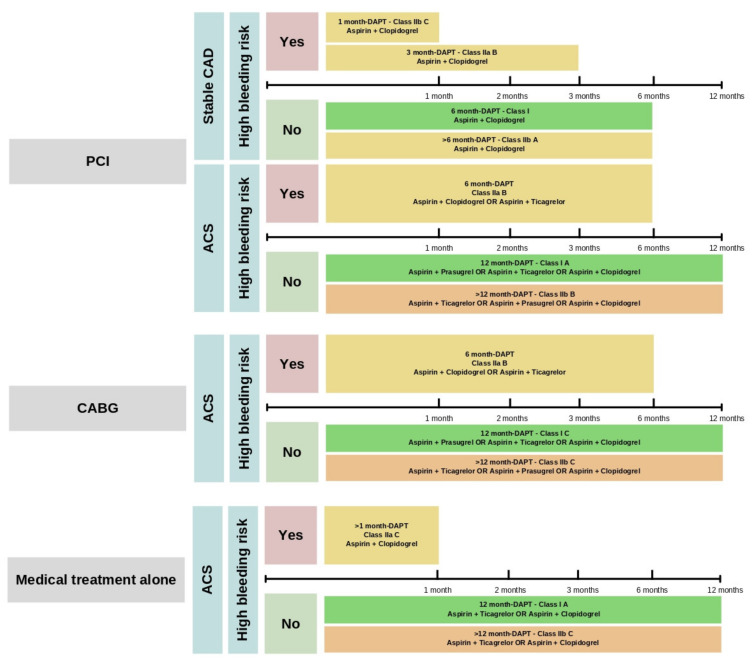
DAPT recommendations according to the intervention approach and bleeding risk. DAPT is not indicated for patients with stable CAD undergoing medical therapy alone or CABG as a treatment option. Abbreviations: ACS: Acute Coronary Syndrome; CABG: Coronary Artery Bypass Graft; DAPT: Dual Antiplatelet Therapy; PCI: Percutaneous Coronary Intervention; Stable CAD: Stable Coronary Artery Disease.

**Table 1 life-13-01580-t001:** Pharmacological profile of common antiplatelet medications.

Medication	Properties	Dose	Adverse Effects	Contraindications
**Inhibitor of the enzyme cyclooxygenase-1 (COX-1)**				
Aspirin	Antiplatelet, analgesic, antipyretic	75–325 mg/day	Bleeding, gastrointestinal ulcers, tinnitus, Reye’s syndrome	Hypersensitivity, active bleeding, history of bleeding disorders, recent surgery
Indobufen	Antiplatelet	200–300 mg/day	Bleeding, gastrointestinal ulcers, dyspepsia	Hypersensitivity, active bleeding, history of bleeding disorders
**P2Y_12_ receptor inhibitors**				
Clopidogrel	Antiplatelet	75 mg/day	Bleeding, gastrointestinal ulcers, thrombotic thrombocytopenic purpura	Hypersensitivity, active bleeding, history of bleeding disorders
Prasugrel	Antiplatelet	10 mg/day	Bleeding, gastrointestinal ulcers	Hypersensitivity, active bleeding, history of bleeding disorders, previous stroke or transient ischemic attack
Ticagrelor	Antiplatelet	90 mg/twice daily	Bleeding, gastrointestinal ulcers, dyspnea	Hypersensitivity, active bleeding, history of bleeding disorders
**Phosphodiesterase inhibitors**				
Dipyridamole	Antiplatelet, vasodilator	200–400 mg/day	Headache, gastrointestinal upset, hypotension	Hypersensitivity, active bleeding, history of bleeding disorders
Cilostazol	Antiplatelet, vasodilator	100 mg/twice daily	Headache, gastrointestinal upset, hypotension	Heart failure, bleeding disorders, recent myocardial infarction
**Protease-activated receptor-1 antagonists**				
Vorapaxar	Antiplatelet	2.08 mg/day	Bleeding, gastrointestinal ulcers, intracranial hemorrhage	History of stroke, transient ischemic attack, bleeding disorders

## Data Availability

Not applicable.

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
