# Peer review of "Dual Antiplatelet Therapy: A Concise Review for Clinicians"

_life, 2023, doi:10.3390/life13071580_

Round 1

Reviewer 1 Report

Please see detailed comments in the file attached.

English language is fine in general. Some parts of the manuscript require editing in terms of syntax, grammar and writing flow.

Author Response

                                                                                               NYU School of Medicine          

                                                                                               550 First Avenue, New York, NY 10016              

                                                                                               Chayakrit Krittanawong, M.D.

July 4th, 2023

Dear Editor

I hope this letter finds you well. I am writing in response to the reviewers’ comments regarding our manuscript "Dual Antiplatelet Therapy: A Concise Review for Clinicians" recently submitted to Life.

We sincerely appreciate all 4 reviewer's time and effort in evaluating the manuscript. While I acknowledge the reviewer's comments and constructive criticism, several points require clarification and further discussion. In this letter, we address additional comments individually and provide additional information to address these issues adequately. We believe that these revisions significantly enhance the quality and clarity of our manuscript. We remain grateful for the opportunity to publish our work in your journal.

Thank you for your time and consideration. We look forward to your feedback on our revisions and hope for a positive outcome.

Yours sincerely,

Chayakrit Krittanawong, MD

Reviewer #1

The authors of this manuscript entitled ‘‘Dual Antiplatelet Therapy: A State-of-the-art Review’’ discuss the use of dual antiplatelet therapy in several clinical scenarios, which is indeed a never-ending hot topic. The review provided is not exhaustive in length, which would be considered reader-friendly, but then there are parts within the text that are rather confusing, irrelevant, or even sloppy. Below, detailed remarks are presented to the authors.

The introduction section is too short. It should be enriched.

Response: We agree that the point raised is indeed important and warrants further elaboration. We have now expanded on this aspect in the revised manuscript to provide a more comprehensive understanding of the topic. We hope that these additions have effectively addressed your concern.

A figure showing the different mechanisms of action for each antiplatelet agent (along with a comprehensive legend describing the pathways that each agent targets) would significantly improve the quality of the paper, even more so since it is characterized as a state-of-the-art review.

Response: Thank you for your insightful suggestion regarding the need to provide more comprehensive details of our article by including figures. We have incorporated a figure depicting the mechanisms of action for each antiplatelet agent and legends describing the pathways that each agent targets.

A table describing the properties, doses, adverse effects and contraindications of the antiplatelet agents mentioned would be very useful for readers.

Response: Thank you for your valuable suggestion. We have included a table showing the properties, doses, adverse effects, and contraindications for each antiplatelet medication mentioned in this review. We sincerely appreciate your feedback.

In some parts there is lack of coherence in terms of writing flow and meaning:

Line 120: Please correct the space between ‘’hand,’’ and ‘’The’’ and also use lower case letter (the instead of The).

Response: Thank you for your valuable suggestion. We have taken note of the sentence you mentioned and made the necessary correction as per your recommendation. Your input has been instrumental in improving the overall quality and accuracy of the text, and we sincerely appreciate your feedback.

Line 128-129: ‘’A meta-analysis indicated that ASA plus dipyridamole significantly reduced stroke risk compared to ASA alone (RR 0.77, 95% CI 0.67-0.89)’’. Please provide a reference.

Response: Your input has greatly enriched the manuscript, and we appreciate your valuable contribution. We have included the article with reference 29. We sincerely appreciate your feedback.

Line 154: Please correct the space between ‘’period(.)’’ and ‘’The’’

Response: Thank you for your valuable suggestion. We have taken note of the sentence you mentioned and made the necessary correction as per your recommendation. Your input has been instrumental in improving the overall quality and accuracy of the text, and we sincerely appreciate your feedback.

Lines 151-154: ‘’ Since DAPT was utilized initially for a period of 12 months for ACS and the prevention of drug-eluting stent thrombosis, short-term DAPT generally refers to a duration of therapy shorter than this, often for 3-6 months, and some more recent studies suggest just 1 month in high bleeding risk patients.’’ Please provide references.

Response: Your input has greatly enriched the manuscript, and we appreciate your valuable contribution. We have included the article with references 40-41. We sincerely appreciate your feedback.

Lines 155-157: ‘’ In the CURE trial, a randomized control trial, DAPT with clopidogrel and ASA in patients with non-ST-elevation ACS in comparison to placebo, 3-12 months of therapy (mean 9 months) showed beneficial effects’’. Please correct syntax by rephrasing.

Response: Thank you for your valuable suggestion. We have taken note of the sentence you mentioned and made the necessary correction as per your recommendation. Your input has been instrumental in improving the overall quality and accuracy of the text, and we sincerely appreciate your feedback.

Line 161: ‘’The management of non-ST elevation ACS as guided by the 2014 ACC/AHA guidelines,’’. It would be nice to also mention the corresponding 2020 ESC guidelines and make a comparison.

Response: Thank you for your valuable recommendation. We have added a comparison on this topic in the section Evidence for Short-term Dual Antiplatelet Therapy. Your input has been outstanding in improving the overall quality and accuracy of the text, and we sincerely appreciate your feedback.

Line 164: “a increased’’ Please correct.

Response: Thank you for your valuable suggestion. We have taken note of the sentence you mentioned and made the necessary correction as per your recommendation. Your input has been instrumental in improving the overall quality and accuracy of the text, and we sincerely appreciate your feedback.

Line 167: Please correct the space between ‘’PCI,’’ and ‘’short-term’’.

Response: Thank you for your valuable suggestion. We have taken note of the sentence you mentioned and made the necessary correction as per your recommendation. Your input has been instrumental in improving the overall quality and accuracy of the text, and we sincerely appreciate your feedback.

Line 169: ‘’acute coronary syndrome’’. Please use ACS, since it has already been abbreviated and used several times before.

Response: Thank you for your valuable suggestion. We have made the necessary correction as per your recommendation. Your input has been instrumental in improving the overall quality and accuracy of the text, and we sincerely appreciate your feedback.

Line 171: ‘’between three- and 12-month’’. Please use either 3- and 12- or three- and twelve-

Response: Thank you for your valuable recommendation. We have made the necessary correction as per your recommendation. Your input has been instrumental in improving the overall quality and accuracy of the text, and we sincerely appreciate your feedback.

Line 171-173: Here it should also be mentioned that no gender differences were observed in the REDUCE trial.

Response: Thank you for your valuable suggestion. We have made the necessary comments mentioning that no gender differences were seen in the REDUCE trial. We sincerely appreciate your feedback.

Line 178: ‘’ did not demonstrate noninferiority’’. I am not sure whether readers will get confused with the use of two negatives and lose the meaning of the sentence.

Response: We appreciate your insightful suggestion. We reformulate the sentence to make sure readers understand. Thank you again for your advice.

Lines 174-181: The inclusion of STOPDAPT-2 ACS under the paragraph ‘’ Evidence for Short-term Dual Antiplatelet Therapy’’ is erroneous. It should be included under the heading ‘’ Evidence for Extended-Duration Dual Antiplatelet therapy’’

Response: We value your insightful suggestion. We moved this section from “Evidence for Short-term Dual Antiplatelet Therapy” to “Evidence for Extended-Duration Dual Antiplatelet therapy”. We are grateful for your suggestions to improve the quality of our manuscript.

Lines 182-183: ‘’In the SMART-CHOICE trial, patients received aspirin + P2Y12 inhibitor for 3 months or P2Y12 inhibitor alone or DAPT for 12 months.’’ Please rephrase, because the sentence is a bit confusing regarding randomization. One may think that patients received from the beginning either DAPT for 3 months or P2Y12 inhibitor alone for 12 months or DAPT for 12 months, whereas they all received aspirin + P2Y12 inhibitor for 3 months and thereafter they were assigned to either P2Y12 inhibitor alone or DAPT for 12 months.

Response: We sincerely appreciate your insightful comments, as they have helped me refine and strengthen my arguments. We have reformulated the paragraph to make sure no confusion is present.

Line 186: Please correct the space between ‘’months.’’ and ‘’extended’’. Also correct the space in Lines 353 and 380.

Response: We have made the necessary correction as per your recommendation. Your expertise and guidance have been invaluable throughout the review process, and we are sincerely grateful for your support.

Line 191: ‘’ with an additional several years, but not more, of therapy’’. Please rephrase.

Response: We sincerely appreciate your insightful comments, as they have helped me refine and strengthen my arguments. We have reformulated the paragraph to make sure no confusion is present.

Line 199: Please omit the period between ‘’thienopyridine’’ and ‘’(43)’’.

Response: We have made the necessary correction as per your recommendation. Your expertise and guidance have been invaluable throughout the review process, and we are sincerely grateful for your support.

Line 200: Please omit the comma in the phrase ‘’multicenter, trial’’.

Response: We have made the necessary correction as per your recommendation. Your expertise and guidance have been invaluable throughout the review process, and we are sincerely grateful for your support.

Line 213: Please replace ‘’ with a median duration of 28 months’’ with ‘’within a median duration of 28 months’’.

Response: We have made the necessary correction as per your recommendation. Your expertise and guidance have been invaluable throughout the review process, and we are sincerely grateful for your support.

Lines 227-228: Please correct the sentence ‘’ Ticagrelor monotherapy for 23 months compared to standard DAPT for 12 months, followed by ASA monotherapy for 12 months’’, which makes no sense. Apart from correcting the syntax, you should specifically describe the two arms of the study design so that the reader can actually understand what each arm includes. At the end, please include the appropriate reference of the study. The one you included is not the GLOBAL-LEADERS study, but a comment to it.

Response: We sincerely appreciate your insightful comments, as they have helped me refine and strengthen my arguments. We have reformulated the paragraph to make sure no confusion is present. We have included the proper reference for this paragraph. Thank you for your valuable suggestions.

Line 230: Please omit the period between ‘’groups’’ and ‘’(49)’’.

Response: We have made the necessary correction as per your recommendation. Your expertise and guidance have been invaluable throughout the review process, and we are sincerely grateful for your support.

Lines 234-239: ‘’According to the 2018 ESC Guidelines, AF patients with DES should receive DAPT for at least 6 months. They can then continue with a single antiplatelet agent or DAPT for up to 12 months or longer based on individual patient characteristics and bleeding risk. For AF patients with Bare Metal Stent (BMS), the recommended duration is 1 month of DAPT followed by up to 6 months of single antiplatelet therapy or DAPT, depending on individual factors (50).’’ I am not sure if these guidelines specifically provide recommendation based on the use of BMS or DES. Could you point out the specific part of the guidelines where this is addressed? Moreover, according to reference 53 (p.193e) ‘’ The 2017 ESC Focused Update on Antiplatelet Therapy recommends that choice of duration of DAPT in patients should no longer be differentiated on basis of device used, i.e. whether the stent implanted at time of PCI is a DES or bare-metal stent, or whether a drug eluting balloon is used.’’ Could you also address this issue and modify your text accordingly?

Response: Thank you for your valuable suggestion. In response to your feedback, we have reformulated the information regarding the issue you have mentioned. We appreciate your guidance in enhancing the clarity and completeness of the paper. References 56 and 57.

Lines 239-242: ‘’The AHA/ACC Focused Update suggests 12 months of DAPT for DES patients, followed by a single antiplatelet agent or DAPT, and at least one month of DAPT for BMS patients, followed by 6 months of single antiplatelet therapy or DAPT (51).’’ The authors should specify if this sentence refers to Afib patients or not. If not, it is irrelevant in this part of the paragraph which talks about Afib patients. Similarly, reference 51 should be replaced with a more appropriate one, since it is stated in this focused update that ‘’ The recommended management of patients on “triple therapy” (aspirin, P2Y12 inhibitor, and oral anticoagulant) is beyond the scope of this focused update. ‘’. Altogether, this sentence and reference should be placed in a different and more appropriate part of the manuscript to avoid any confusions.

Response: Thank you for your valuable suggestion. In response to your feedback, we deleted this section and reformulated the paragraph using the reference 56. We appreciate your guidance in enhancing the clarity and completeness of the paper.

Lines 254-257: There is lack of coherence with the subtitle and the previous sentences. The subtitle reads ‘’DURATION OF ANTIPLATELET THERAPY POST-CABG’’ and the previous sentences start an introduction about DAPT after CABG. Instead of analyzing the subject further, the authors, immediately after these introductory sentences, start analyzing DAPT before CABG.

Response: Thank you for your valuable suggestion. In response to your feedback, we deleted this section and reformulated the paragraph. We appreciate your guidance in enhancing the clarity and completeness of the paper.

Lines 260-265: Reference 44 is completely irrelevant to the sentences included in these lines. Also, why did the authors put bold in these lines?

Response: Your input has greatly enriched the manuscript, and we appreciate your valuable contribution. We have included a reference supporting this paragraph. You can find the article with reference 57.

Lines 267-268: Reference 56 does not support this sentence, since its subject is not about TAVI. Instead, it talks about aortic stenosis in general and contains the 2002 recommendations for surgical aortic valve replacement.

Response: Your input has greatly enriched the manuscript, and we appreciate your valuable contribution. We have included a reference supporting this paragraph. You can find the article with reference 59.

Lines 274-276: ‘’Afterward, switch to oral anticoagulation monotherapy. For patients not needing anticoagulation, use ASA or clopidogrel during the periprocedural period.’’ The writing style should be consistent all throughout the text. These sentences use imperative form and are more like instructions, thus diverging from the appropriate writing style of the previous and following sentences.

Response: Thank you for your valuable recommendation. We have made the necessary correction as per your recommendation. Your input has been instrumental in improving the overall quality and accuracy of the text, and we sincerely appreciate your feedback.

Lines 288-291: Either reference 60 or the sentence in these lines should be revised. Otherwise, the authors need to specify the part of the cited article clearly supporting this sentence. Also, the authors should rephrase the following phrase ‘’for the prevention of atherothrombotic events in secondary prevention’’

Response: Your input has greatly enriched the manuscript, and we appreciate your valuable contribution. We have reformulated this paragraph. With regard to the cited article (reference 64), you can find in the abstract section those findings written in our manuscript. Your input has been instrumental in improving the overall quality and accuracy of the text, and we sincerely appreciate your feedback.

Lines 292-294: ‘’A systematic review of walking distance trials found that antiplatelet therapy for 6 months was significantly superior to placebo in enhancing the distance walked without experiencing extremity discomfort with a mean difference of 78 feet (61) (62) (63) (64).’’ Here the reference should be the systematic review mentioned. References 61-63 are not systematic reviews. Reference 64 is the only systematic review among the 4 cited. However, reference 64 examined the effectiveness of antiplatelet agents in reducing mortality and cardiovascular events, and not differences in walking distance (at least based on the information provided in its abstract). Could the authors please specify whether reference 64 is the appropriate reference?

Response: Thank you for your valuable suggestion. In response to your feedback, we deleted this section. We appreciate your guidance in enhancing the clarity and completeness of the paper.

Line 297: (65) (66) (67) (65) (68) Reference 65 is written twice. So, there are actually 4 references, but the authors mention five trials in Line 295. Which one is the fifth?

Response: Thank you for your insightful recommendation. In response to your feedback, we deleted the duplicated reference and mentioned “Various meta-analyses have shown that cilostazol is effective in treating stable moderate-to-severe claudication(65)(66)(67)(68)” to support our comment. We appreciate your guidance in enhancing the clarity and completeness of the paper.

Lines 299-303: Please check if the percentages mentioned in parentheses are correct based on reference 47. Your input has been instrumental in improving the overall quality and accuracy of the text, and we sincerely appreciate your feedback.

Response: We appreciate your comment. We ensured that the percentages mentioned in the parenthesis are correct.

Line 307: Please remove the period in the following ‘’or drug-eluting). (73)’’

Response: Thank you for your valuable recommendation. We have made the necessary correction as per your recommendation. Your input has been instrumental in improving the overall quality and accuracy of the text, and we sincerely appreciate your feedback.

Line 308-310: ‘’ The Zilver PTX study compared DES to angioplasty in patients with femoropopliteal artery disease. DES showed superior results in survival, patency, and freedom from reintervention compared to standard care ’’ Please specify ‘’angioplasty’’ and ‘’standard care’’.

Lines 307-322: All these sentences are irrelevant with the subtitle ‘’DURATION OF ANTIPLATELET THERAPY IN PAD POST-PERIPHERAL STENT’’. Instead of analyzing antiplatelet therapy, the authors discuss here the use of DES, DCB and PTA.

Response: Thank you for your comment. We have reformulated the section and added new information. Please, review the reference 73, 75, 76, supporting our paragraphs. We appreciate your guidance in enhancing the clarity and completeness of the paper.

Lines 331-332: ‘’with symptomatic intracranial symptomatic stenosis’’. Please correct.

Response: Thank you for your insightful suggestion. We have taken your recommendation into consideration and made the necessary revisions to the sentence. We appreciate your guidance in refining the clarity and formal language of the message.

Lines 332-333: ‘’ the ESC guidelines recommends’’. Please correct.

Response: Thank you for your insightful suggestion. We have taken your recommendation into consideration and made the necessary revisions to the sentence. We appreciate your guidance in refining the clarity and formal language of the message.

Line 364 (disease), Line 366 (for patient), Line 378 (the risk revascularization)

Response: Thank you for your insightful suggestion. We have taken your recommendation into consideration and made the necessary revisions to the sentence. We appreciate your guidance in refining the clarity and formal language of the message.

Figure 1: Please put numbers 18 and 24 in the correct position above the arrow. Also include CABG in the abbreviations. Use the same font in all the words of the figure. Please correct ‘’on the duration of the dual antiplatelet therapy’’.

Response: Thank you for your insightful suggestion. We have removed the figure. We agreed that the previous figure needed to be more accurate. We decided to create two new figures exhibiting the action mechanisms of each antiplatelet medication (Figure 1) and legends describing the pathways each agent targets, , one showing DAPT recommendations according to the intervention approach and bleeding risk like those published in ESC guidelines (Figure 2). Your input has been valuable in improving the clarity and comprehensiveness of the discussion. We greatly appreciate your contribution.

Table 1: Please put CABG, MI and DAPT in the abbreviations. Also correct ‘’ In most clinical settings is made for at least 6–12 months of DAPT. A recommendation is made for prolonged DAPT beyond this initial 6- to 12-month period’’ and ‘’ In patients with prior MI at high ischemic risk under medical therapy alone and have tolerated DAPT’’

Response: Thank you for your insightful suggestion. We have removed the table. We agreed that the previous table needed to be more accurate. We added a table showing the properties, doses, adverse effects, and contraindications for each antiplatelet medication mentioned in this review. Your input has been valuable in improving the clarity and comprehensiveness of the discussion. We greatly appreciate your contribution.

Reference 55 is not written properly. The paper is available on pubmed as follows:             Sandner S, Redfors B, Angiolillo DJ, Audisio K, Fremes SE, Janssen PWA, Kulik A, Mehran R, Peper J, Ruel M, Saw J, Soletti GJ, Starovoytov A, Ten Berg JM, Willemsen LM, Zhao Q, Zhu Y, Gaudino M. Association of Dual Antiplatelet Therapy With Ticagrelor With Vein Graft Failure After Coronary Artery Bypass Graft Surgery: A Systematic Review and Meta-analysis. JAMA. 2022 Aug 9;328(6):554-562. doi: 10.1001/jama.2022.11966.

Response: Thank you for your valuable recommendation. We have deleted this reference from the manuscript. Your input has been useful in enhancing the overall quality and precision of the text, and we sincerely appreciate your feedback.

Reference 92: Please correct null null.

Response: Thank you for your valuable recommendation. We have deleted this reference from the manuscript. Your input has been useful in enhancing the overall quality and precision of the text, and we sincerely appreciate your feedback.

In several parts of the manuscript, discussion for each topic is rather limited and sometimes confusing. All major trials for each topic should be discussed first, followed by current guidelines on the subject from major societies and ending with key messages for each topic. I believe that this structure would be useful and friendly to the reader.

Response: Thank you for your insightful suggestion. We have reorganized the different sections accordingly. Your input has been valuable in improving the clarity and comprehensiveness of the discussion. We greatly appreciate your contribution.

References 12 and 13 are the same.

Response: Thank you for your valuable suggestion. We have taken note of the duplicated references. Your input has been instrumental in improving the overall quality and accuracy of the text, and we sincerely appreciate your feedback.

References 42 and 44 have not been included and analyzed in the text. Authors jump from reference 41 directly to reference 43 and from 43 to 45.

Response: Thank you for your valuable recommendation. We have deleted references 42 and 44. Your input has been useful in enhancing the overall quality and precision of the text, and we sincerely appreciate your feedback.

Reference 49 should be modified, since authors need to include the results of the trial instead of the comment. The original trial is Vranckx P. et al. Ticagrelor plus aspirin for 1 month, followed by ticagrelor monotherapy for 23 months vs aspirin plus clopidogrel or ticagrelor for 12 months, followed by aspirin monotherapy for 12 months after implantation of a drug-eluting stent: a multicentre, open-label, randomised superiority trial. Lancet. 2018 Sep 15;392(10151):940-949. doi: 10.1016/S0140-6736(18)31858-0

Response: Your input has greatly enriched the manuscript, and we appreciate your valuable contribution. We have corrected the reference. You can find the article with reference 54.

When using abbreviations and acronyms, they should first be presented in the expanded form and abbreviated thereafter. e.g. ASA (Line 33), ADP (Line 95), MI (Line 98), PCI (Line 84; has already been explained and abbreviated in Line 51), cAMP (Line 135; has already been abbreviated in Line 125), FDA (Line 147). DES (Line 167), AF (Line 231, 233), ESC (Line 234), AHA/ACC/SCAI (Line 243), CABG (Lines 251,252), TAVR (Line 279), CD-TLR (Line 317), TIA (Line 335). This should be done separately for the abstract and for the main text.

Response: Thank you for your valuable suggestion. We have considered it and made the necessary corrections on this issue in the article. Your input has greatly contributed to enhancing the overall quality and clarity of the article.

I would recommend not using the phrase ‘’heart attack’’.

Response: Thank you for your valuable recommendation. We have deleted the phrase “heart attack”. Your input has been useful in enhancing the overall quality and precision of the text, and we sincerely appreciate your feedback.

Reviewer #2

This article was written well, easy to read, and enjoyable for readers of this journal.

There is one minor comment.

Among TAVR patients, a study comparing clopidogrel monotherapy to aspirin monotherapy reported a reduction of cardiovascular death during the 2-year follow-up regardless of anticoagulation use (Y. Kobari et al. Circulation: Cardiovascular Interventions. 2021;14:e010097).  This article would add insightful information to this article. 

Response: Your input has greatly enriched the manuscript, and we appreciate your valuable contribution. We have included the article “Aspirin Versus Clopidogrel as Single Antithrombotic Therapy After Transcatheter Aortic Valve Replacement: Insight From the OCEAN-TAVI Registry” in the section “Duration of antiplatelet therapy in patients post TAVI”. You can find the article with reference 62.

Reviewer #3

The concept of the article is ok, a synthesis of DAPT, however the data are not new. The text is pertinent, but the figure and the table are not informative, they are too simple for the complexity of the subject. I would like to have for all major indications a separate, more detailed figure, without references on figure, depicting the major modalities/alternatives of DAPT and the criteria for choosing a certain path (similar to the figures from the ESC Guidelines). The table comparing the ESC and AHA/ACC recommendations is not exact, for instance in stable CAD (chronic coronary syndrome) there is no need for DAPT in the majority of cases, it has to be emphasized which category of patients benefits from DAPT, etc. The table has to be modified substantially.   

Response: Thank you for your insightful suggestion. We have removed the figure and table. We agreed that the previous figure and table needed to be more accurate. We decided to create two new figures exhibiting the action mechanisms of each antiplatelet medication (Figure 1) and legends describing the pathways each agent targets, one showing DAPT recommendations according to the intervention approach and bleeding risk like those published in ESC guidelines (Figure 2). Additionally, we added a table showing the properties, doses, adverse effects, and contraindications for each antiplatelet medication mentioned in this review. Your input has been valuable in improving the clarity and comprehensiveness of the discussion. We greatly appreciate your contribution.

Reviewer #4

Authors do not address the seminal COMPASS study that studied the addition of rivaroxaban in combination with aspirin among patients with the chronic coronary syndrome. This critical reference should be added: Eikelboom et al. N Engl J Med. 2017.

Response: Your input has greatly enriched the manuscript, and we appreciate your valuable contribution. We have included the COMPASS trial in the section “Duration of antiplatelet therapy in stable cad” I. You can find the article with reference 55.

I think that this paper should be enhanced with the paragraph tackling an important issue in contemporary antiplatelet management and that is the use of monotherapy regimens in the ACS. This is an important field and should be discussed. There should be a paragraph summarizing the most important studies in this field and the rationale for monotherapy with antiplatelet agents. This review currently lacks that aspect.

Response: Thank you for your valuable suggestion. As you are aware, with the multitude of papers available, we have made a conscientious effort to select those relevant to DAPT clinical applications in the settings proposed in this review. Your input has been instrumental in refining the selection process of information regarding the intention of this manuscript. We greatly appreciate your direction. 

More emphasis should be put on the precision tailoring of antiplatelet therapy - a greater degree of discussion dedicated to balancing ischemic and bleeding risks for specific groups of patients.

Response: Thank you for your valuable suggestion. In response to your feedback, we have incorporated additional information regarding the benefits and risks of DAPT in the introduction section. We appreciate your guidance in enhancing the clarity and completeness of the paper.

Special scenarios of using antiplatelet therapy in cardiovascular disease such as spontaneous coronary artery dissection (SCAD) and MINOCA should be briefly covered - these areas are fields of active research; however, these are fairly common clinical scenarios.

Response: Thank you for your insightful suggestion. We have created sections for both SCAD and MINOCA and the benefits of patients taking DAPT for these scenarios. Your input has been valuable in improving the clarity and comprehensiveness of the discussion. We greatly appreciate your contribution.

Reviewer 2 Report

This article was written well, easy to read, and enjoyable to readers of this journal.

There is one minor comment.

Among TAVR patients, a study comparing clopidogrel monotherapy to aspirin monotherapy reported a reduction of cardiovascular death during the 2-year follow-up regardless of anticoagulation use (Y. Kobari et al. Circulation: Cardiovascular Interventions. 2021;14:e010097).  This article would add insightful information to this article. 

Author Response

(The authors gave the same response as above.)

Reviewer 3 Report

The concept of the article is ok, a synthesis of DAPT, however the data are not new. The text is pertinent, but the figure and the table are not informative, they are too simple for the complexity of the subject. I would like to have for all major indications a separate, more detailed figure, without references on figure, depicting the major mdalities/alternatives of DAPT and the criteria of choosing a certain path (similar to the figures from the ESC Guidelines). The table comparing the ESC and AHA/ACC recommendations is not exact, for instance in stable CAD (chronic coronary syndrome) there is no need for DAPT in the majority of cases, it has to be emhasized which category of patients benefits from DAPT, etc. The table has to be modified substantially.   

Author Response

(The authors gave the same response as above.)

Reviewer 4 Report

Dear Editor and authors, I had the pleasure to read this in-depth and generally-well written review covering the use of dual antiplatelet therapy for the management of cardiovascular diseases.

I feel there are certain areas of this manuscript that should be covered in order to improve its quality, as outlined below:

- Authors do not address the seminal COMPASS study that studied the addition of rivaroxaban in combination with aspirin among patients with the chronic coronary syndrome. This critical reference should be added: Eikelboom et al. N Engl J Med. 2017.

- I think that this paper should be enhanced with the paragraph tackling an important issue in contemporary antiplatelet management and that is the use of monotherapy regimens in the ACS. This is an important field and should be discussed. There should be a paragraph summarizing the most important studies in this field and the rationale for monotherapy with antiplatelet agents. This review currently lacks that aspect.

- More emphasis should be put on the precision tailoring of antiplatelet therapy - a greater degree of discussion dedicated to balancing ischemic and bleeding risks for specific groups of patients.

- Special scenarios of using antiplatelet therapy in cardiovascular disease such as spontaneous coronary artery dissection (SCAD) and MINOCA should be briefly covered - these areas are fields of active research, however, these are fairly common clinical scenarios.

Author Response

(The authors gave the same response as above.)

Round 2

Reviewer 3 Report

The manuscript became more clear and comprehensive after adjustments. I would like to have an other figure depicting DAPT in the other clinical scenarios presented in the text (PAD, TAVI, carotid disease, etc.).

none

Reviewer 4 Report

Thank you for addressing my comments. No further questions.

No concerns.